# Intracellular cGMP increase is not involved in thyroid cancer cell death

Sara D'Alessandro[1,2], Elia Paradiso[1], Clara Lazzaretti[1], Samantha Sperduti[1,3], Carmela Perri[1], Francesco Antoniani[1], Sara Righi[1], Manuela Simoni[1,3,4], Giulia Brigante[1,4], Livio Casarini [1,3]*

1 Unit of Endocrinology, Department of Biomedical, Metabolic and Neural Sciences, University of Modena and Reggio Emilia, Modena, Italy, 2 International PhD School in Clinical and Experimental Medicine (CEM), University of Modena and Reggio Emilia, Modena, Italy, 3 Center for Genomic Research, University of Modena and Reggio Emilia, Modena, Italy, 4 Unit of Endocrinology, Department of Medical Specialties, Azienda Ospedaliero-Universitaria di Modena, Modena, Italy

* livio.casarini@unimore.it

## Abstract

### Introduction

Type 5 phosphodiesterase (PDE5) inhibitors (PDE5i) lead to intracellular cyclic-guanosine monophosphate (cGMP) increase and are used for clinical treatment of erectile dysfunction. Studies found that cGMP may up/downregulate the growth of certain endocrine tumor cells, suggesting that PDE5i could impact cancer risk.

### Aim

We evaluated if PDE5i may modulate thyroid cancer cell growth *in vitro*.

### Materials and methods

We used malignant (K1) and benign (Nthy-ori 3–1) thyroid cell lines, as well as the COS7 cells as a reference model. Cells were treated 0–24 h with the PDE5i vardenafil or the cGMP analog 8-br-cGMP (nM-μM range). cGMP levels and caspase 3 cleavage were evaluated by BRET, in cGMP or caspase 3 biosensor-expressing cells. Phosphorylation of the proliferation-associated extracellularly-regulated kinases 1 and 2 (ERK1/2) was evaluated by Western blotting, while nuclear fragmentation by DAPI staining. Cell viability was investigated using 3-(4,5-dimethylthiazol-2-yl)-2,5-diphenyltetrazolium bromide (MTT) assay.

### Results

Both vardenafil and 8-br-cGMP effectively induced dose-dependent cGMP BRET signals (p≤0.05) in all the cell lines. However, no differences in caspase 3 activation occurred comparing PDE5i-treated *vs* untreated cells, at all concentrations and time-points tested (p>0.05). These results match those obtained upon cell treatment with 8-br-cGMP, which failed in inducing caspase 3 cleavage in all the cell lines (p>0.05). Moreover, they reflect the lack of nuclear fragmentation. Interestingly, the modulation of intracellular cGMP levels

**Data availability statement:** All source files and raw data are available from the Figshare database (doi: 10.6084/m9.figshare.22151912).

**Funding:** Italian Ministry of University and Research supported the Department of Biomedical, Metabolic and Neural Sciences (University of Modena and Reggio Emilia, Italy) in the context of the Departments of Excellence Programme. No specific authors were funded. Grant number is not available and the Funder did not play any role in the study design, data collection and analysis, decision to publish, or preparation of the manuscript.

**Competing interests:** The authors have declared that no competing interests exist.

with vardenafil or the analog did not impact cell viability of both malignant and benign thyroid tumor cell lines, nor the phosphorylation of ERK1/2 (p>0.05).

## Conclusions

This study demonstrates that increased cGMP levels are not linked to cell viability or death in K1 and Nthy-ori 3–1 cell lines, suggesting that PDE5i do not impact the growth of thyroid cancer cells. Since different results were previously published, further investigations are recommended to clarify the impact of PDE5i on thyroid cancer cells.

## Introduction

The 3',5'-cyclic guanosine monophosphate (cGMP) is a second messenger activating the nitric oxide (NO)-dependent soluble guanylyl cyclase (sGC) or the natriuretic peptides (NPs)-dependent particulate guanylyl cyclase (pGC). They have a cGMP-elevating effect, activating second messenger-dependent protein kinases G (PKG), phosphodiesterases (PDEs) and ion channels. Dysregulation of these intracellular signaling pathways may lead to pathological effects, since cGMP is involved in cell proliferation, as well as in cancer development and progression [1]. Its ability to modulate cell proliferation and apoptosis depends on the cell or tumor type, which differently responds to up/down-regulation of NO/cGMP pathway [1–3]. For this reason, it is hard to predict how drugs interfering with cGMP cascade, such as PDE5 inhibitors (PDE5i), could impact cancer cell proliferation.

PDE5i were first introduced for the treatment of erectile dysfunction (ED). Vardenafil is one of these synthetic and PDE5-specific compounds that competitively inhibits the hydrolysis of cGMP, thus, increasing the intracellular levels of the second messenger [4]. To date, these kind of drugs are clinically useful for the treatment of pulmonary hypertension, cardiovascular disease, and of benign prostatic hyperplasia (BHP) symptoms [3, 5]. PDE5i are also used to enhance effects of chemotherapeutic medications. In fact, certain tumor cells develop resistance to these drugs over time, due to a molecular mechanism based on ATP-binding cassette (ABC) transponders overexpression [6]. It results in the drug exporting out from the cell, lowering the therapeutic efficacy. In this case, PDE5i acts as a non-toxic transporter modulator and enhancer of drug efficacy [3].

Several studies characterized the cancer cell-specific effect of cGMP pathway, which is linked to proliferation or death. For instance, increasing cGMP levels may upregulate apoptotic events and inhibit tumorigenesis of colon cancer cells [7–10], breast cancer cells [8, 11, 12], gastric cancer cells [8, 13, 14], pancreatic adenocarcinoma cells [15] leukemia and myeloma [3], prostate carcinoma [16] and glioma cells [8]. Moreover, cGMP/PKGII signaling was found to exert inhibitory effects on growth and metastasis in various human tumor cell lines *in vitro*, including lung, hepatic, renal, colon cancer and glioma cell lines [8]. On the contrary, anti-apoptotic effects or increased risk of tumor development were found respectively in ovary carcinoma cells [17] and melanoma [3].

The link between intracellular cGMP levels and thyroid disorders was hypothesized several decades ago [18–20]. This issue was investigated in thyroid nodules of animal models, where the second messenger could modulate the production of inflammatory mediators [21]. Most importantly, studies in human biopsies revealed that increased cGMP levels may trigger autophagic signals counteracting proliferation of papillary thyroid cancer (PTC) cells and preventing their expansion to other tissues and organs [22]. Similar conclusions were achieved

by bioinformatic approaches, suggesting that downregulation of cGMP pathway-related genes may be associated with PTC [23]. However, data about the role of cGMP in PTC, as well as if PDE5i may modulate thyroid tumor cell growth are relatively scarce.

In this *in vitro* study, we explored whether intracellular cGMP increase may be a target for PTC therapy. Intracellular levels of the second messenger were intentionally perturbed by adding a cGMP analog or a PDE5i to cell media, and pro/anti-apoptotic signals were evaluated. To this purpose, the human PTC cell line K1 was used as a model of thyroid malignant cancer, while the follicular epithelial cell line Nthy-ori 3–1 as a benign cell of thyroid origin.

## Materials and methods

Source files and raw data are available in an online repository (doi: 10.6084/m9.figshare.22151912).

### Reagents

8-br-labeled GMP (8-Br-cGMP) (CAS.N.:51116-01-9, Santa Cruz Biotechnology, Santa Cruz, CA, USA) and vardenafil hydrochloride CRS (CAS.N.: 330808-88-3, Merck KGaA) were used to treat cells (nM-μM concentration range) and induce the increase of intracellular cGMP levels. 4 μM thapsigargin (Tocris Bioscience, Bristol, UK) was used as a positive control in caspase 3 activation analysis, nuclear fragmentation and 3-(4,5-dimethylthiazol-2-yl)-2,5-diphenyltetrazolium bromide (MTT) assay. 0.1 μM phorbol myristate acetate (PMA) (Sigma-Aldrich) and 10 μM of the mitogen-activated protein kinase kinase (MEK) inhibitor U0126 (Sigma-Aldrich) were used as positive and negative controls for ERK1/2 phosphorylation, respectively [24]. 10 μM NS 2028 (CAS.N.: 204326-43-2, Merck KGaA) was used as inhibitor of sGC.

### Cultured cell lines

The K1 cell line was used as a model of malignant thyroid tumor, i.e. PTC [25]. The Nthy-ori 3–1 cell line were from immortalized epithelial thyroid cells [26] and served as a model of healthy thyroid. The COS7 cell line was used as a reference cell line and were available in-house. K1 cells were cultured with Dulbecco's modified eagle medium (DMEM), Ham's F12 and MCDB 105 (all from Sigma-Aldrich, a division of Merck KGaA, Darmstadt, Germany) in a 2:1:1 ratio. Nthy-ori 3–1 cells were grown in RPMI-1640, while COS7 were cultured with DMEM (Sigma-Aldrich). All these cell media were enriched by 10% FBS, 2 mM L-glutamine, 100 U/ml penicillin and 50 μg/ml streptomycin (all from Sigma-Aldrich).

### Plasmids and transfection

The C163A-pcw107-V5 Caspase 3 BRET biosensor [27] was obtained by Addgene (Watertown, MA, USA) and validated in the present study. GFP$^2$-GAFa-Rluc sensor was previously described [28] and validated [29]. Transient transfections of cell lines were performed in a 96-well plate using Metafectene PRO (Biontex Laboratories GmbH, Munich, Germany), following the manufacturer's protocol. To this purpose, 50 ng and 200 ng of plasmid coding caspase 3 and cGMP biosensors, respectively, were added to each well after to be mixed with 0.5 μl of Metafectene PRO diluted in FBS-free medium. A volume of 50 μl plasmid-Metafectene PRO mix was added to each well, containing $15.0 \times 10^3$ cells, achieving a total volume of 250 μl/well. Cells were incubated 2 days and starved overnight with FBS-free medium before treatments with 8-Br-cGMP and vardenafil.

## BRET measurements

Caspase 3 cleavage was evaluated in COS7, K1 and Nthy-ori 3–1 cell lines transiently expressing the specific (C163A)-pcw107-V5 BRET biosensor. In this molecule, a luciferase NanoLuc (NLuc) is fused to the fluorescent acceptor protein mNeonGreen (mNG) via a caspase-cleavable 17 amino acid flexible linker. In the presence of caspase 3, the linker undergoes cleavage, leading to dissociation between NLuc and mNG and resulting in the decrease of BRET signal. cGMP intracellular increase was measured in cell lines transiently expressing the $GFP^2$-GAFa-Rluc biosensor [29, 30]. Binding between cGMP and the specific GAF domain results in biosensor conformational changes, leading to BRET signal increase. Reactions occurred in the presence of the luciferase enzyme substrate coelenterazine H (NanoLight Technologies, a division of Prolume Ltd., Lakeside, AZ, USA). The cGMP and caspase 3 BRET biosensors detection range was evaluated by dose-response experiments using 8-Br-cGMP and thapsigargin serial dilutions, respectively, after 2 and 4 hours of treatment. Cells were also treated with 8-br-cGMP 1 μM and Vardenafil 50 μM in the presence/absence of Guanylyl cyclase inhibitor 10 μM. 8-br-cGMP 20 μM and Thapsigargin 4 μM were respectively used as positive control for cGMP and caspase3 study to validate the detection range of the sensor. The analysis of cell death signals was extended to pro-apoptotic gene expression (S1 File).

## pERK1/2 protein analysis

Proliferative signals were investigated using ERK1/2 phosphorylation, as previously described [31, 32]. To these purposes, cell lines were seeded in 24-well plates ($1.0x10^5$ cells/well) and treated 15 min with 1 μM 8-Br-cGMP or 50 μM vardenafil. Where needed, cells were pre-treated 1 h with the NS 2028 GC inhibitor, PMA or U0126 [24]. Cells were lysed in ice-cold RIPA buffer containing protease and phosphatase inhibitors for protein extractions. The phosphorylation of ERK1/2 was evaluated using an anti-pERK1/2 primary antibody (#9101, Cell Signaling Technology Inc., Danvers, MA, USA). Lysed samples were loaded to separate proteins in a 10% sodium dodecyl sulfate (SDS)-polyacrylamide gel electrophoresis and transferred to a polyvinylidene fluoride membrane. The anti-total ERK antibody (#9102, Cell Signaling Technology Inc.) was used as a loading control. Then, membranes were incubated with a secondary anti-rabbit horseradish peroxidase (HRP)-conjugated antibody (#NA9340V; GE HealthCare).

Western blotting signals were revealed upon incubation with ECL chemiluminescent compound (GE HealthCare, Chicago, IL, USA). Images were acquired by the VersaDoc™ MP 4000 System and QuantityOne analysis software (Bio-Rad Laboratories Inc., Hercules, CA, USA).

## Cell viability assay

Cell viability was investigated by MTT assay [33], using a known procedure [31]. Cells were seeded in 96-wells plates ($1.0x10^4$ cells/well) and serum-starved over-night before 24-h treatments with increasing concentrations of 8-Br-cGMP (100 nM-50 μM) and vardenafil (100 nM-500 μM). 100 μl/well of assay solution (#M5655; Sigma-Aldrich) was prepared by dissolving 1 mg of MTT powder into 1 ml of cell medium. After 2-h incubation at 37°C, in the presence of 5% of $CO_2$, the solution was removed and 100 μl/well of isopropanol was added to dissolve the formazan crystals produced by metabolically active cells. Absorbance was detected at the wavelength of 570 nm by a Victor3 plate reader (Perkin Elmer Inc., Waltham, MA, USA).

## Nuclear fragmentation

To investigate whether the intracellular cGMP increase impacts thyroid cancer cell death, $1.5 \times 10^4$ cells/well were seeded in 24-well plates and treated 4–24 h with 1 μM 8-Br-cGMP or 50 μM vardenafil. Thapsigargin served as a positive control inducing nuclear fragmentation [34, 35]. Cells were fixed by 10% formalin and DAPI-stained 20 min. Images were captured by a fluorescence microscope (Nikon Eclipse Ts2R; Nikon Corporation, Tokyo, Japan).

## Statistical analysis

Statistical analysis and graphic representation were obtained by GraphPad Prism 9.0 (Graph-Pad Prism Software Inc., La Jolla, CA, USA) Results were analyzed by Kruskal Wallis test and Dunn's post-test, after to have applied the D'Agostino and Pearson normality test. Differences were considered significant for $p < 0.05$.

# Results

## Intracellular cGMP dose-finding

The optimal 8-Br-cGMP and vardenafil concentration to be used *in vitro* was evaluated by dose-finding experiments in transfected COS7, K1 and Nthy-ori 3–1 cell lines expressing the cGMP BRET biosensor. Cells were treated 2 h with increasing 8-Br-cGMP and vardenafil concentrations before signal detection by BRET. In COS7 and K1 cell lines, the lowest 8-Br-cGMP concentration resulting in detectable signals was 1 μM, while it was 50 μM for vardenafil (Kruskal Wallis test and Dunn's post-test; $p < 0.05$). Similar effects were found with 500 μM 8-Br-cGMP in the Nthy-ori 3–1 cell line ($p < 0.05$), while vardenafil failed in inducing detectable cGMP increase in this model (Fig 1). cGMP detection by the BRET biosensor was further investigated under cell co-treatment with 10 μM of the sGC inhibitor NS 2028 and 8-Br-cGMP/vardenafil (S1 Fig), confirming the lack of signal in the absence of cGMP.

These experiments served to identify the optimal PDE5i and cGMP analog concentration range to be used in further experiments, where we evaluated if the modulation of intracellular cGMP levels impacts cell viability and death *in vitro*. To this purpose, the 100 nM– 500 μM 8-Br-cGMP/vardenafil range was chosen as the most indicative of cGMP increase. The effective, 1 mM concentration was excluded because it was unnecessarily high to obtain intracellular cGMP increase and because it was near to the vardenafil solubility.

## Intracellular increase of cGMP does not affect cell viability

The viability of COS7, K1 and Nthy-ori 3–1 cells was evaluated under 8-Br-cGMP and vardenafil treatment. To this purpose, cells were treated 24 h with the PDE5i or the analog and viability was evaluated by MTT assay. Both the compounds did not reduce cell viability at all the concentration tested (Fig 2), except 500 μM vardenafil (Kruskal Wallis test and Dunn's post-test; $p < 0.05$). In this case, results could be biased by sub-optimal dissolution of the compound in the vehicle, which resulted in an opaque solution since the concentration used dropped around the solubility of compound in water [36]. Control experiments confirming that cGMP perturbation does not impact cell viability were performed under sGC blockade by NS 2028 (S2 Fig).

Analysis of the cell viability was deepened by evaluating ERK1/2 phosphorylation, as a hallmark of cell survival [37]. To this purpose, cells treated with 8-Bromo-cGMP and vardenafil were used for Western blotting analysis and pictures were semi-quantified to be represented in graphs (Fig 3). Cells treated with PMA and the MEK inhibitor U0126 served as positive and negative controls, respectively. We found that both 8-Bromo-cGMP and vardenafil failed

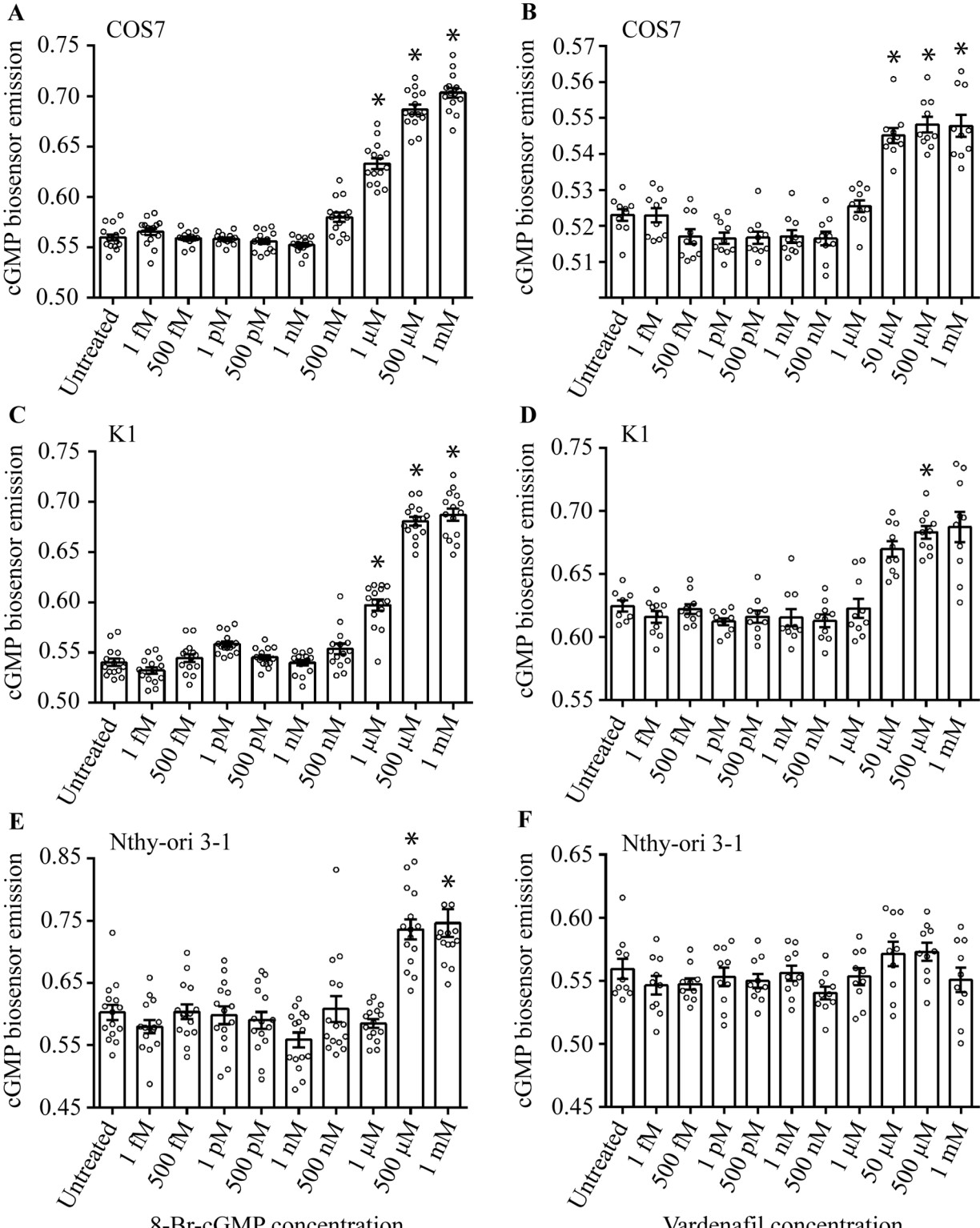

**Fig 1. Analysis of intracellular cGMP levels.** Transfected cells expressing the cGMP/Rluc-tagged biosensor were treated 2 h with increasing 8-Br-cGMP and vardenafil (fM-mM range). BRET signals indicating intracellular cGMP levels were evaluated and plotted against the concentration of inhibitor/analog. A, B) Data from the COS7 cell line treated with 8-Br-cGMP and vardenafil, respectively; C, D) 8-Br-cGMP- and vardenafil-treated K1 cell line; E, F) Nthy-ori 3–1 cells treated with 8-Br-cGMP and vardenafil. Results were compared by Kruskal Wallis test and Dunn's post-test *versus* untreated samples and statistically significant differences were marked by * (p<0.05; means ± SEM; n = 8 to 15).

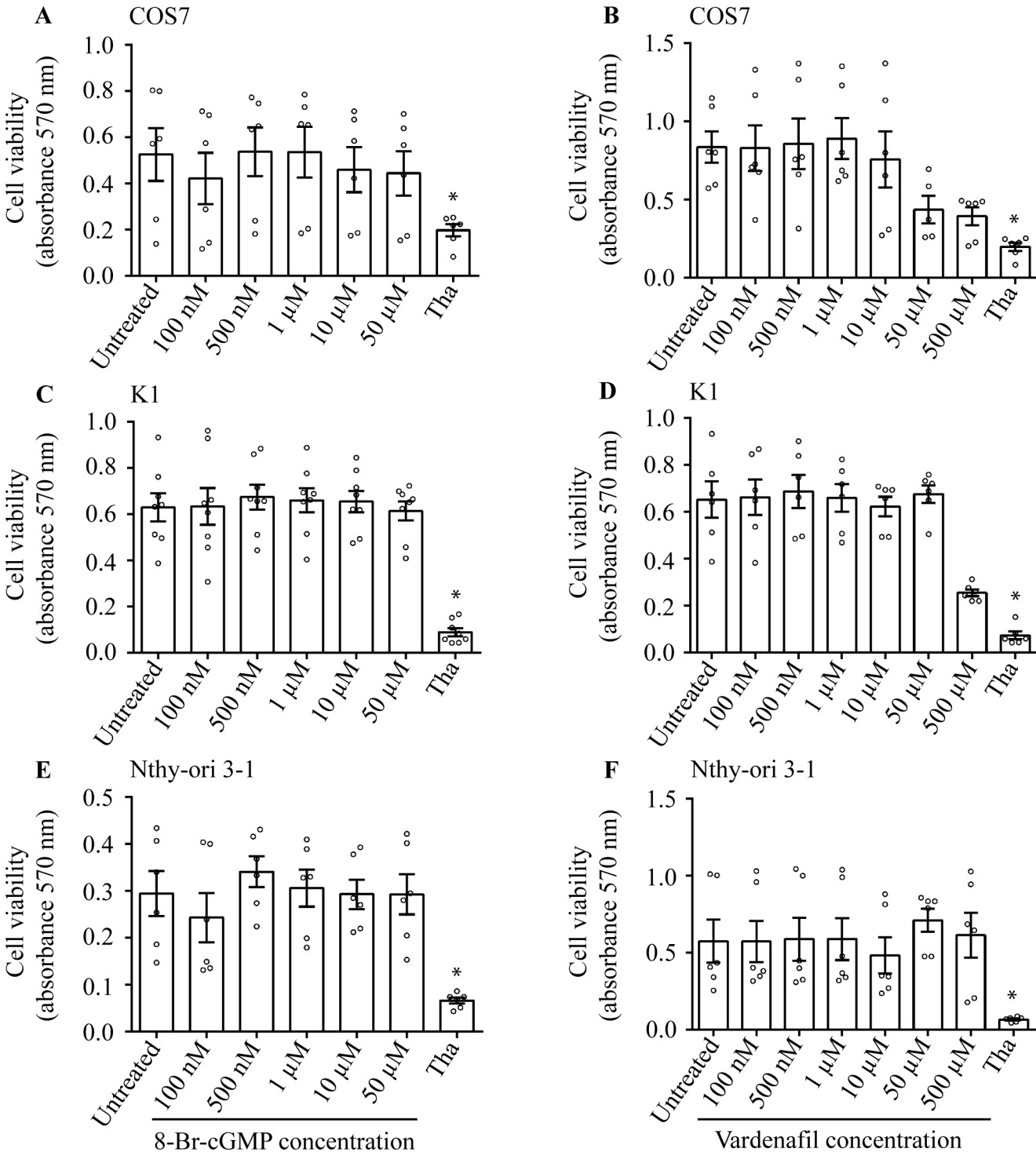

**Fig 2. Modulation of cell viability by 8-Br-cGMP/vardenafil.** COS7, K1 and Nthy-ori 3–1 cell lines were treated 24 h with increasing 8-Br-cGMP and vardenafil (100 nM– 500 μM range). Cell viability was evaluated by MTT assay and indicated as an absorbance detected at the 570 nm wavelength. 4 μM thapsigargin served as a positive control (Tha) for cell viability decrease and results were compared by Kruskal Wallis test and Dunn's post-test (*significantly different *versus* untreated sample; p < 0.05; means ± SEM; n = 6 to 8). A, B) Data from the COS7 cell line treated with 8-Br-cGMP and vardenafil, respectively; C, D) Results from treatments of K1 cells; E, F) Nthy-ori 3–1 cell line.

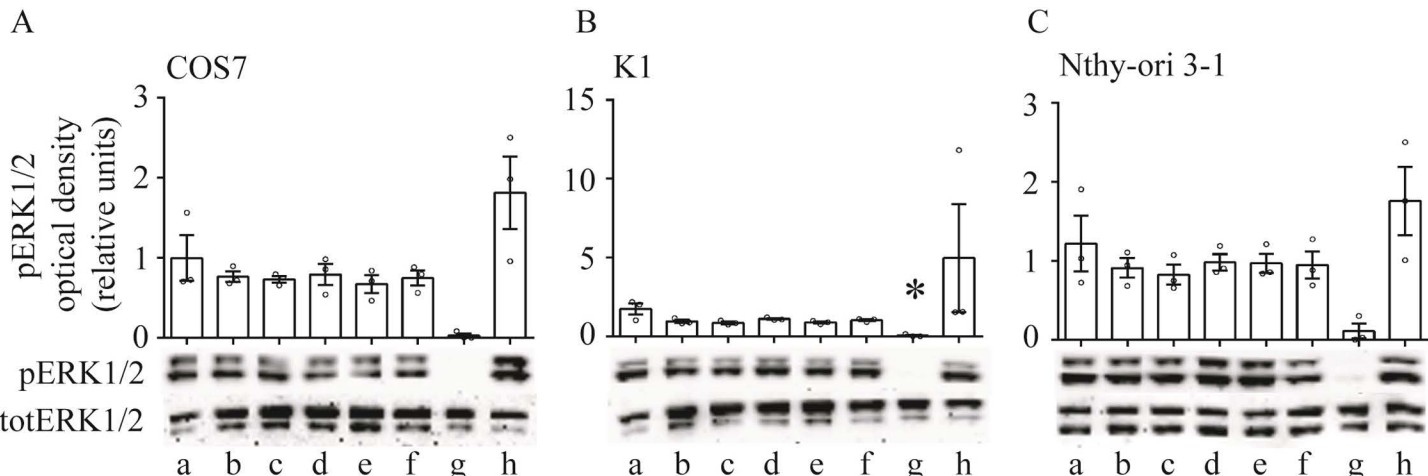

**Fig 3. 15-min, 8-Br-cGMP/vardenafil-induced ERK1/2 phosphorylation in the three cell lines.** Cells were treated as follow: a) untreated; b) 10 μM 8-Br-cGMP; c) 50 μM vardenafil; d) preincubation with NS 2028 (GCi) and treatment with 8-Br-cGMP; e) preincubation with NS 2028 (GCi) and treatment with vardenafil; f) NS 2028 (GCi); g) MEK inhibitor U0126 was used to deplete ERK1/2 phosphorylation h) PMA was used as positive control of pERK1/2. Data from densitometric analysis were compared by Kruskal Wallis test and Dunn's post-test (*significantly different *versus* untreated samples; $p \geq 0.05$; means ± SEM were superimposed to data points; n = 3).

in modulating pERK1/2 activation compared to the basal level detected in untreated samples (Kruskal Wallis test and Dunn's post-test; $p \geq 0.05$).

## 8-Bromo-cGMP and vardenafil weakly impact caspase 3 activity

Caspase 3 activation was evaluated in cell lines treated 4 h with the cGMP analog and PDE5i, by BRET. To this purpose, the expression of a specific biosensor for the cleaved caspase 3 activity was induced upon vector transfection. Thapsigargin was used as a positive control [38, 39] We found that both 8-Br-cGMP and vardenafil slightly reduced the caspase 3 activity in COS7 cells, while the activity of the enzyme was potently enhanced upon thapsigargin treatment as in all the other tested cell lines. However, in K1 cells, almost all the concentrations tested of both the cGMP analog and PDE5i induced only a weak increase of the caspase 3 activity, detected as a significantly higher BRET signal (Fig 4; Kruskal Wallis test and Dunn's post-test; $p < 0.05$). Similar results were obtained in Nthy-ori 3–1 cells treated with 100 nM– 1 μM 8-Br-cGMP ($p < 0.05$), while vardenafil induced no effects on caspase 3 activity. The overall weak effect of cGMP perturbation on caspase 3 activity, if any, was confirmed by control experiments using the sGC inhibitor (S3 Fig), and further confirmed by the lack of pro-apoptotic *TP53* gene expression (S4 Fig).

Analysis of caspase 3 activity was followed by evaluations of nuclear damage in cell lines treated 24 h with 1 μM 8-Br-cGMP and 50 μM vardenafil. Cell nuclei were stained by DAPI and pictures captured by a fluorescent microscope. Consistently with the weak or no activation of caspase 3 activity, both treatments with cGMP analog and PDE5i did not induce nuclear fragmentation (Fig 5), and oppositely to what obtained upon cell incubation with thapsigargin (positive control).

## Discussion

This study demonstrates that the perturbation of intracellular cGMP levels did not affect pro/anti-apoptotic signals in two *in vitro* models representative of malignant and benign thyroid cells, as well as in the COS7 reference cell line. Both the second messenger analog 8-Br-cGMP and the PDE5i vardenafil failed to induce cell death signals, e.g. consistent caspase 3 activity

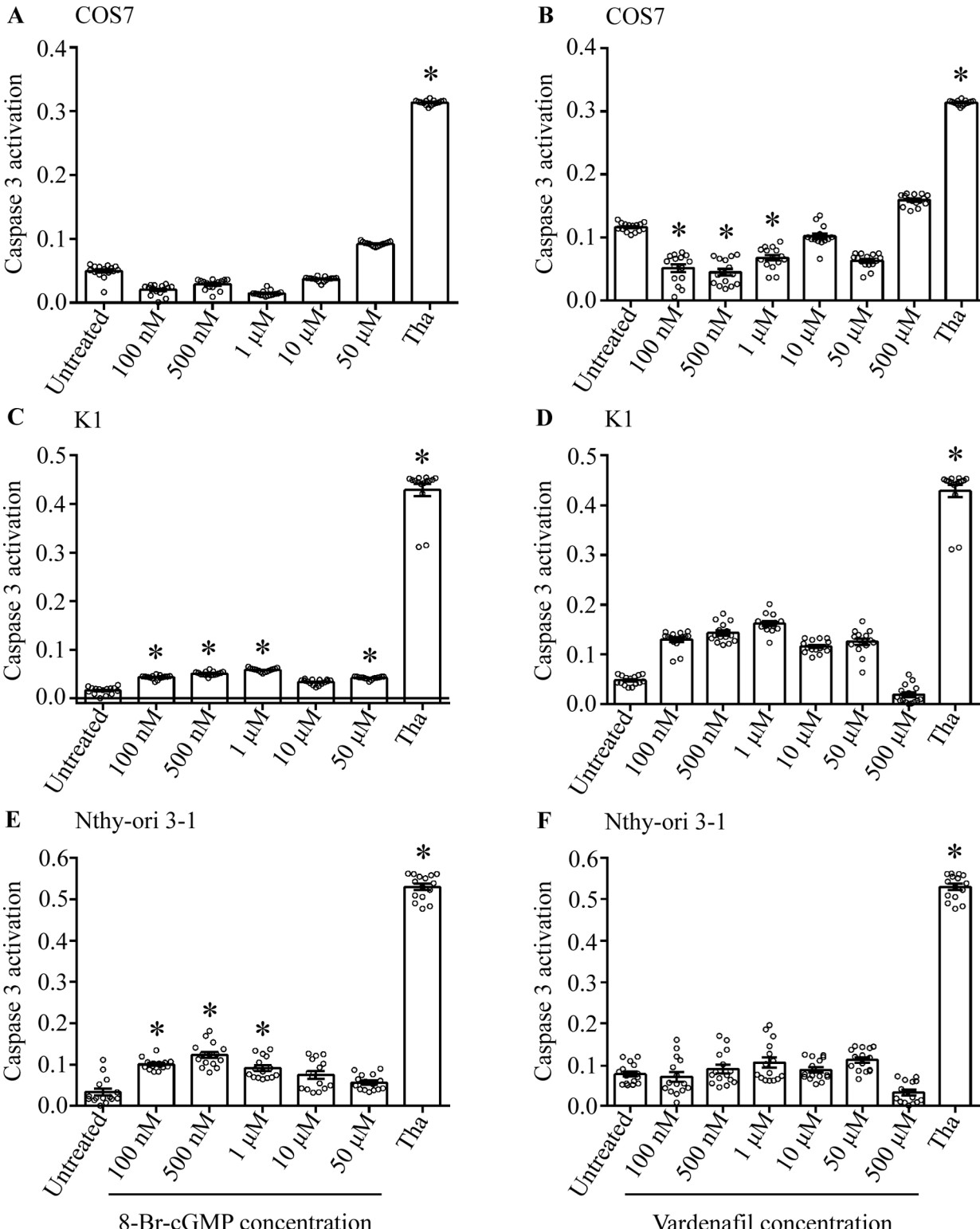

**Fig 4. Caspase 3 activity in 8-Br-cGMP/vardenafil-treated COS7, K1 and Nthy-ori 3–1 cell lines.** The activity of the enzyme was detected by BRET, in cells treated 4 h with increasing 8-Br-cGMP and vardenafil concentrations (100 nM– 500 μM range) and expressing a specific biosensor. 4 μM thapsigargin served as a positive control (Tha). Results were compared by Kruskal Wallis test and Dunn's post-test (*significantly different *versus* untreated sample; p < 0.05; means ± SEM; n = 15). A, B) 8-Br-cGMP- and vardenafil-treated COS7 cells, respectively; C, D) Results from K1 cells; E, F) Nthy-ori 3–1 cell line.

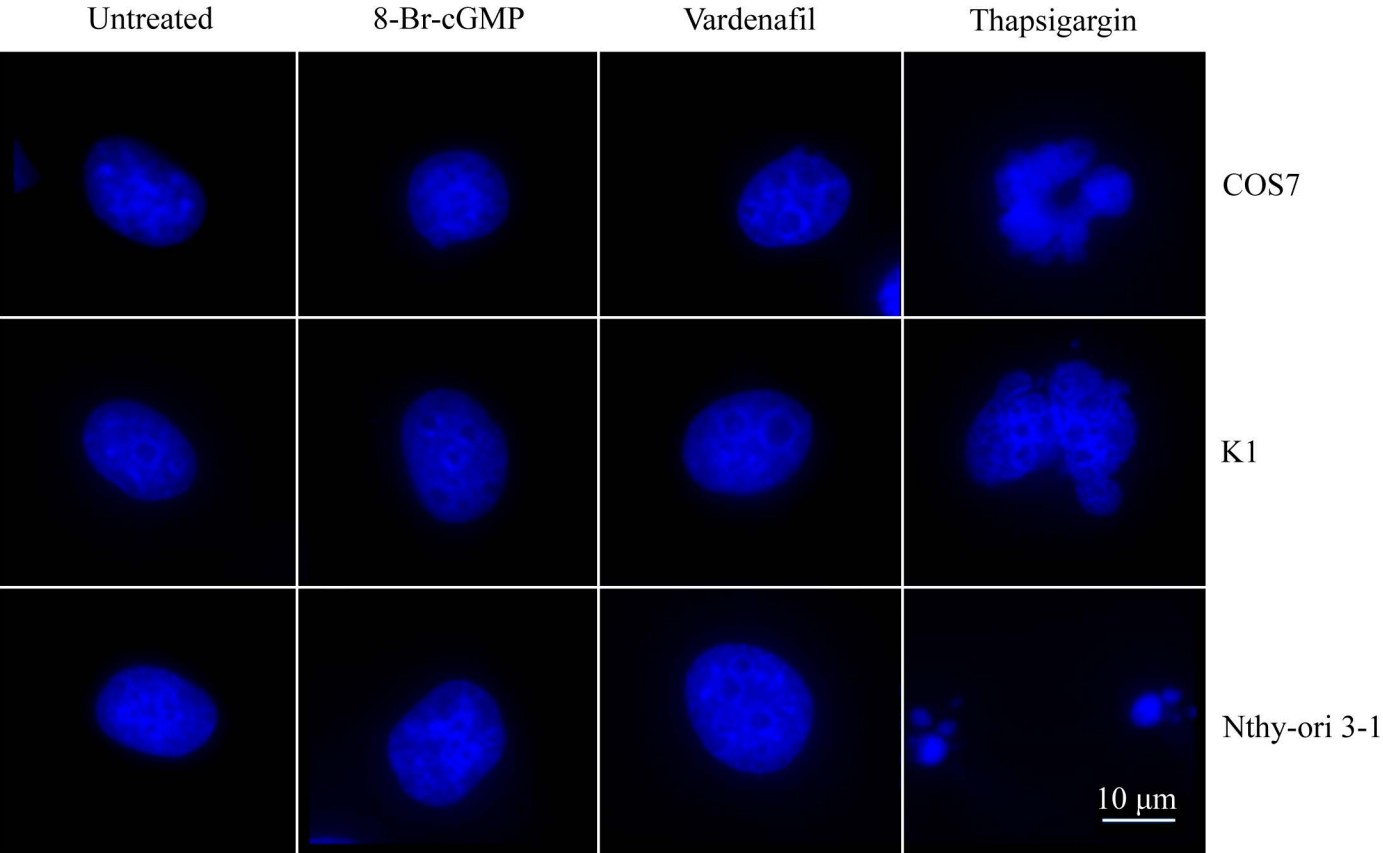

**Fig 5. Evaluation of nuclear damage.** COS7, K1 and Nthy-ori 3–1 cells were treated 24 h with 1 μM 8-Br-cGMP and 50 μM vardenafil before to be fixed and DAPI stained. 4 μM thapsigargin-treated cells served as positive controls. 60x magnification; scale bar = 10 μM; n = 1.

and nuclear damage. These data match the failure of sGC inhibitor in enhancing/inhibiting cell viability, confirming a weak effect of cGMP, if any, in the modulation of thyroid cell line metabolism. On one hand, our results suggest that cGMP could not be a target for PTC therapy. On the other hand, they indicate that PDE5i would not impact the growth of malignant thyroid cells and might be used for clinical treatment of erectile dysfunction in PTC patients.

The relevance on this study is found in the increasing use of PDE5i [40], as well as in the dual role of cGMP in tumor cells, where the second messenger may upregulate growth and proliferation or, on the contrary, exert pro-apoptotic effects, depending on the kind of tumor [3, 41]. For instance, cGMP suppresses apoptosis and upregulates proliferative signals in several cancer cells, such as ovarian [17], breast [42], renal carcinoma [43] and head/neck squamous cell carcinoma [44]. For this reason, the use of PDE5i could not be recommended in certain cancer patients. An *in vitro* study demonstrated that the PDE5 inhibition by sildenafil increases the intracellular levels of cGMP which, in turn, potentiates the mitogen-activated protein kinase (MAPK)-dependent melanoma cell growth [45] and increases the risk of tumorigenic effect [46]. However, opposite results were found by studies in different cancer cells and cell lines. cGMP induced apoptosis *via* PKG in certain breast cancer cell lines [11], in colon [47, 48] and gastric cancer cells [49]. Efforts were made to determine why some cancer cells are responsive to cGMP-induced apoptosis. A study suggested that various types of cancer may be selectively sensitive to cGMP-mediated apoptosis, due to a crosstalk

between intracellular pathways linked to the second messenger and death receptors specifically expressed in those cells [50, 51]. Moreover, we may speculate that different responses to cGMP may be due to cell-specific PDE5 expression levels, as well as of genes involved in the regulation of metabolic functions involving the cyclic nucleotide. Therefore, in-depth "omics" analyses should be performed to solve this unanswered question, given that *PDE5A*, *BRAF*, *KRAS*, *HRAS* and *NRAS* gene expression did not differ significantly among cell lines and thyroid cells from benign and malignant nodules (S5 Fig). Taken together, the cancer cell-specific impact of cGMP in regulating life/death signals suggests that studies should be dedicated to focus its role in cells of thyroid origin.

In thyroid tissues from biopsy samples, cGMP signaling was associated with protective mechanisms against PTC invasion [22]. These data may suggest that PDE5i could provide a new approach to treat thyroid cancer, as confirmed by other *in vitro* studies. For instance, the administration of PDE5i-loaded nanovesicles reduced the viability and proliferation of two human papillary thyroid carcinoma cell lines (TPC-1 and BCPAP) [52]. Similar results were found previously in the same cell lines, as well as in the human papillary thyroid carcinoma cell line 8505C [53]. These data were strengthened by the detection of higher PDE5 expression levels in PTC than normal thyroid tissue [53], reflecting the anti-proliferative effect of cGMP accumulation and PDE5i in thyroid cancer cells.

Although some findings may propose PDE5i as promising tools for PTC, the lack of consistent amount of data and the existence of opposite results suggest being cautious. First, the aforementioned studies evaluated the effect of PDE5i different than vardenafil, i.e. sildenafil and tadalafil, which share the same enzyme as a main target, but demonstrated different binding affinities [54, 55], and targeting of other PDEs [30, 56]. Since the PDE5 is not the unique cGMP-specific enzyme [57], these differences between PDE5i may explain tissue- and cell-specific effects. Indeed, we found no modulation of K1 and Nthy-ori 3–1 thyroid cell line viability by vardenafil, neither within μM concentrations of the compound, confirming the existence of drug- and cell-specific effects. Other explanations may rely on cGMP-independent activation of PKG, which was demonstrated using synthetic peptides [58] and occurring naturally in certain tissues [59, 60]. Interestingly, PDE5 inhibition by the drug may occur via different, cGMP-dependent and -independent intracellular mechanism. PDE5 has two highly homologous, distinct regulatory domains, GAF-A and GAF-B, which have different affinity to the inhibitor [61]. In human platelet, the sensitivity of PDE5 for sildenafil inhibition was modulated upon blockade of cGMP-binding sites of the high-affinity GAF-A domain, leading to preferential PDE5i binding to GAF-B [61] suggestive of non-canonical effects at the intracellular level.

## Conclusions

We found that intracellular cGMP modulation by vardenafil does not impact the viability of benign and malignant thyroid cell models *in vitro*. While these data may suggest that the clinical use of PDE5i for erectile dysfunction could not impact the growth of thyroid cancer cells, poor but opposite results reflect the extremely cell-specific nature of the drug effect on tumors. This issue must be further investigated.

## Supporting information

**S1 Fig.**
(TIF)

**S2 Fig.**
(TIF)

**S3 Fig.**
(TIF)

**S4 Fig.**
(TIF)

**S5 Fig.**
(TIF)

**S1 File.**
(DOCX)

## Acknowledgments

Authors are grateful to the Italian Ministry of University and Research.

## Author contributions

**Conceptualization:** Manuela Simoni, Giulia Brigante, Livio Casarini.

**Data curation:** Sara D'Alessandro, Elia Paradiso, Clara Lazzaretti, Samantha Sperduti, Carmela Perri.

**Formal analysis:** Sara D'Alessandro, Elia Paradiso, Clara Lazzaretti, Samantha Sperduti, Carmela Perri, Francesco Antoniani, Sara Righi.

**Investigation:** Sara D'Alessandro, Elia Paradiso, Clara Lazzaretti, Samantha Sperduti, Carmela Perri, Francesco Antoniani, Sara Righi, Giulia Brigante.

**Methodology:** Giulia Brigante, Livio Casarini.

**Project administration:** Livio Casarini.

**Supervision:** Manuela Simoni, Giulia Brigante, Livio Casarini.

**Validation:** Giulia Brigante, Livio Casarini.

**Visualization:** Livio Casarini.

**Writing – original draft:** Sara D'Alessandro, Livio Casarini.

**Writing – review & editing:** Manuela Simoni, Giulia Brigante, Livio Casarini.

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
