## [Decision Letter · Decision Letter 0]

12 Jan 2023

PONE-D-22-29667cGMP is not involved in thyroid cancer cell deathPLOS ONE

Dear Dr. Casarini,

Thank you for submitting your manuscript to PLOS ONE. After careful consideration, we feel that it has merit but does not fully meet PLOS ONE’s publication criteria as it currently stands. Therefore, we invite you to submit a revised version of the manuscript that addresses the points raised during the review process.

We look forward to receiving your revised manuscript.

Kind regards,

Avaniyapuram Kannan Murugan, M.Phil., Ph.D.

Academic Editor

PLOS ONE

Journal Requirements:

2. Please clarify from where you obtained the cell lines used in your study.

“University of Modena and Reggio Emilia, Modena, Italy, and the Italian Ministry of University and Research.”

“Italian Ministry of University and Research supported the Department of Biomedical, Metabolic and Neural Sciences (University of Modena and Reggio Emilia, Italy) in the context of the Departments of Excellence Programme. No specific authors were funded. Grant number is not available and the Funder did not play any role in the study design, data collection and analysis, decision to publish, or preparation of the manuscript.”

7. PLOS ONE now requires that authors provide the original uncropped and unadjusted images underlying all blot or gel results reported in a submission’s figures or Supporting Information files. This policy and the journal’s other requirements for blot/gel reporting and figure preparation are described in detail at https://journals.plos.org/plosone/s/figures#loc-blot-and-gel-reporting-requirements and https://journals.plos.org/plosone/s/figures#loc-preparing-figures-from-image-files. When you submit your revised manuscript, please ensure that your figures adhere fully to these guidelines and provide the original underlying images for all blot or gel data reported in your submission. See the following link for instructions on providing the original image data: https://journals.plos.org/plosone/s/figures#loc-original-images-for-blots-and-gels.

Reviewers' comments:

Reviewer's Responses to Questions

**Comments to the Author**

1. Is the manuscript technically sound, and do the data support the conclusions?

Reviewer #1: Yes

Reviewer #2: Partly

Reviewer #3: Yes

2. Has the statistical analysis been performed appropriately and rigorously?

Reviewer #1: Yes

Reviewer #2: Yes

Reviewer #3: Yes

3. Have the authors made all data underlying the findings in their manuscript fully available?

Reviewer #1: Yes

Reviewer #2: Yes

Reviewer #3: Yes

4. Is the manuscript presented in an intelligible fashion and written in standard English?

Reviewer #1: Yes

Reviewer #2: Yes

Reviewer #3: Yes

5. Review Comments to the Author

Reviewer #1: It might be a good idea for the follow-up study to repeat the experiments adding PPC-1, BCPAP and 8505c cell lines to make sure the discrepancies you found are not due to some "technical issues". Moreover, using an extended panel of the cell lines could be instrumental in order to focus on the cell-specific nature of the drug effect on tumors

Reviewer #2: In this study, the authors attempted to clarify the role of PDE5i and cGMP in thyroid cancer cell biology in vitro. The story of the paper provides some theoretical basis for the safety of PDE5i treatment for erectile dysfunction patients with thyroid cancer, it is of some clinical value, and considerably simple and easy to be understood. However, due to the simpleness, the story should be more rigorous. As commented below, there are not a few compromised points in this manuscript. Therefore, in my view,this manuscript in its current form, is unsuitable for Journal of PLOS ONE.

1)The title of the paper is not strict. If they only show results using PDE5i inhibitors or cGMP analogues which lead to the elevation of cGMP, “Increased cGMP is not involved in thyroid cancer cell death” may be more suitable. It should be more directly linked to the contents of the paper.

2)The PDE5 activity assay and PDE5 mRNA level should be tested to confirm the effect of PDE5i vardenafil in thyroid cancer cells.

3) Currently, there are inadequate data to make any conclusive statements “cGMP is not involved in thyroid cancer cell death” by only utilizing the PDE5i and cGMP analog. Specific inhibition (knockdown) of PDF5 using RNA interference should be performed to confirm the role of PDE5 inhibition on thyroid cancer cell.

4) Have the authors investigated expression of PDE5 and cGMP in thyroid cancer tissues or cells compared with normal or other cancer cell lines/tissues that have been reported to be impacted by PDE5i. These types of experiments would significantly enhance the clinical significance of the data. Is there a possibility that the expression level of PDE5 is relatively low or absent in thyroid cancer, thus, further inhibition would unlikely create a greater functional effect. The impact of PDE5 overexpression on thyroid cancer cell biology is required to address.

5) In the supplementary figure 1, one more set using vardenafil should be designed.

Reviewer #3: D’Alessandro and colleagues present a detailed cellular analysis of cGMP modulation in cell lines including kidney fibroblast, normal thyroid and thyroid cancer. Through thapsigargin control and vardenafil stimulation known to induce cGMP pathway activation in penile, prostatic and cardiac cells, the authors convincingly show that cGMP levels are unchanged by biosensor emission without effects on viability. In complementary studies, the authors show that different from literature reports, PDE5 inhibitor vardenafil did not impact ERK phosphorylation by Western blotting, and weakly impacts caspase 3 activity by as specific biosensor and BRET. The authors acknowledge the limitations of the study in terms of vardenafil use as unique agent with the PDE5 drug class, include appropriate negative and positive controls in each experiment. The title is provocative, but supported by the conclusions and adds to the thyroid cancer literature in particular.

-Discussion of specific areas for improvement

--Major Issues:

The readership would be interested in the effect of cGMP modulation on known targets of differentiated papillary thyroid cancer treatment such as RAS or BRAF. The authors partially address BRAF signaling in their pERK1/2 studies presented in Figure 3. Did the authors explore or reference RAS or BRAF activation or protein expression across their cell line panel? If not, would suggest discussing this as a limitation of the study.

--Minor Issues

*As a reviewer I appreciate including the number of data points comprising each bar graph. For clarity, I would suggest mentioning the n of biological samples in the figure captions. This is done inconsistently through the figures, for instance mentioned in the caption of Figure 2 but not 1, 3 or 4.

*The caption of figure 2 reports n=3, but the data points per bar number close to 6 in panels A-D. Can the authors clarify in the captions or methods this discrepancy? Did the authors perform 3 cellular experiments per condition with duplicates?

6. PLOS authors have the option to publish the peer review history of their article (what does this mean?). If published, this will include your full peer review and any attached files.

Reviewer #1: No

Reviewer #2: No

Reviewer #3: No

---

## [Author Response · Author response to Decision Letter 0]

23 Feb 2023

Reply to reviewers and editors has been provided as a separate file.

---

## [Decision Letter · Decision Letter 1]

21 Mar 2023

Intracellular cGMP increase is not involved in thyroid cancer cell death

PONE-D-22-29667R1

Dear Dr. Casarini,

We’re pleased to inform you that your manuscript has been judged scientifically suitable for publication and will be formally accepted for publication once it meets all outstanding technical requirements.

An invoice for payment will follow shortly after the formal acceptance. To ensure an efficient process, please log into Editorial Manager at http://www.editorialmanager.com/pone/ , click the 'Update My Information' link at the top of the page, and double check that your user information is up-to-date. If you have any billing related questions, please contact our Author Billing department directly at authorbilling@plos.org.

Kind regards,

Avaniyapuram Kannan Murugan, M.Phil., Ph.D.

Academic Editor

PLOS ONE

Additional Editor Comments:

Reviewers' comments:

Reviewer's Responses to Questions

**Comments to the Author**

1. If the authors have adequately addressed your comments raised in a previous round of review and you feel that this manuscript is now acceptable for publication, you may indicate that here to bypass the “Comments to the Author” section, enter your conflict of interest statement in the “Confidential to Editor” section, and submit your "Accept" recommendation.

Reviewer #1: All comments have been addressed

Reviewer #2: All comments have been addressed

Reviewer #3: All comments have been addressed

2. Is the manuscript technically sound, and do the data support the conclusions?

Reviewer #1: Yes

Reviewer #2: Yes

Reviewer #3: Yes

3. Has the statistical analysis been performed appropriately and rigorously?

Reviewer #1: Yes

Reviewer #2: Yes

Reviewer #3: Yes

4. Have the authors made all data underlying the findings in their manuscript fully available?

Reviewer #1: Yes

Reviewer #2: Yes

Reviewer #3: Yes

5. Is the manuscript presented in an intelligible fashion and written in standard English?

Reviewer #1: Yes

Reviewer #2: Yes

Reviewer #3: Yes

6. Review Comments to the Author

Reviewer #1: (No Response)

Reviewer #2: In this study, the authors have identified role of PDE5i and cGMP in thyroid cancer cell apoptosis in vitro. The story is of some clinical value, and considerably simple and easy to be understood. All the comments have been answered reasonable, and there are no additional comments.

Reviewer #3: (No Response)

7. PLOS authors have the option to publish the peer review history of their article (what does this mean?). If published, this will include your full peer review and any attached files.

Reviewer #1: No

Reviewer #2: No

Reviewer #3: No

---

## [Editor Report · Acceptance letter]

22 Mar 2023

PONE-D-22-29667R1

Intracellular cGMP increase is not involved in thyroid cancer cell death

Dear Dr. Casarini:

I'm pleased to inform you that your manuscript has been deemed suitable for publication in PLOS ONE. Congratulations! Your manuscript is now with our production department.

Kind regards,

on behalf of

Dr. Avaniyapuram Kannan Murugan

Academic Editor

PLOS ONE